# Focal lamina cribrosa defects and significant peripapillary choroidal thinning in patients with unilateral branch retinal vein occlusion

Hae Min Kang[1]*, Eun Woo Kim[2], Jeong Hoon Choi[3], Hyoung Jun Koh[2], Sung Chul Lee[2]

1 Department of Ophthalmology, Catholic Kwandong University College of Medicine, International St. Mary's Hospital, Incheon, Republic of Korea, 2 Department of Ophthalmology, Yonsei University College of Medicine, Seoul, Republic of Korea, 3 Choikang Seoul Eye Clinic, Seoul, Republic of Korea

* liebe05@naver.com

## Abstract

### Purpose

To investigate the prevalence of focal lamina cribrosa (LC) defects in patients with unilateral branch retinal vein occlusion (BRVO) and to compare ocular characteristics between eyes with and without focal LC defect and those eyes with normal-tension glaucoma (NTG).

### Methods

This retrospective, cross-sectional study included 121 patients. Thirty-nine patients had unilateral BRVO (BRVO group), 36 patients had NTG (NTG group), and 36 patients had vitreous floaters, but no other ocular diseases (control group). In addition to baseline characteristics such as age, sex, refractive errors, the ocular characteristics such as peripapillary choroidal thickness (PCT), retinal nerve fiber layer thickness, and subfoveal choroidal thickness were retrospectively analyzed.

### Results

Focal LC defects were detected in 20 eyes of 14 patients (38.9%) in the BRVO group, 24 eyes of 15 patients (41.7%) in the NTG group, and none in the control group (P<0.001). In the BRVO-affected eyes, the mean PCT was 102.7±31.1 μm in the eyes with focal LC defects, and 163.1±70.1 μm in the eyes without LC defects (P = 0.009). In the BRVO-affected eyes, the mean PCT was 102.7±31.1 μm in the eyes with focal LC defects, and 163.1±70.1 μm in the eyes without LC defects (P = 0.009). In the NTG group, the mean PCT was 133.1±48.9 μm in the eyes with focal LC defects and 170.8±81.9 μm in those without (P = 0.042). The other baseline and ocular characteristics were not significantly different between the eyes with and without focal LC defects in both the BRVO group and the NTG group.

**Data Availability Statement:** Data cannot be shared publicly because of decision by the Institutional Review Board of International St.

Mary's Hospital. Because the corresponding author (Kang HM) is the only retina specialist, and the patients in this study were all her patients. However, IRB also decided that the data are available from the Ethics Committee (contact to Prof. Je Hooh Park, the head of IRB; ceccil@ish.ac.kr) for researchers who meet the criteria for access to confidential data.

**Funding:** This work was supported by a National Research Foundation of Korea (NRF) grant funded by the Korea government (MSIT) (No. 2018R1C1B5085620). The funder (NRF) did not play a role in the study design, data collection and analysis, decision to publish, or preparation of the manuscript.

**Competing interests:** Choikang Seoul Eye clinic is a private eye clinic, and had no role in the study design, data collection and analysis, decision to publish, or preparation of the manuscript in this study. Choikang Seoul Eye Clinic only provides salaries to Dr.Jeong Hoon Choi. This does not alter our adherence to PLOS ONE policies on sharing data and materials.

## Conclusions

About 40% of the patients with unilateral BRVO had focal LC defect in the BRVO-affected eyes and unaffected fellow eyes, similar prevalence to the patients with NTG. The mean PCT was significantly thinner in the eyes with focal LC defect than those without in the patients with BRVO and those with NTG, suggesting possible pathophysiologic correlation between these two diseases.

## Introduction

Although specific pathophysiological relationships between branch retinal vein occlusion (BRVO) and glaucoma remain under investigation, some studies suggested that significant reductions in the retinal nerve fiber layer (RNFL) thickness occurs in both BRVO-affected eyes and the unaffected fellow eyes.[1,2] Our study group previously investigated whether there are any changes in the peripapillary choroidal thickness (PCT) in patients with unilateral BRVO.[3,4] We found that the mean PCT is significantly reduced over a 12-month period in both BRVO-affected eyes and unaffected fellow eyes.[3,4] The reduction rate was significantly greater in the eyes affected by BRVO when compared with the unaffected fellow eyes.[3,4] Another study found that ocular blood flow is significantly lower in both BRVO-affected eyes and non-affected fellow eyes. This result suggests that significant vascular pathogenesis can occur in eyes of patients with BRVO similar to that described in central retinal vein occlusion and hemi-central retinal vein occlusion.[5]

Subsequent studies have further investigated changes in the optic nerve head (ONH) region, especially the lamina cribrosa (LC), in patients with BRVO.[6,7] These studies found that significant reductions in LC thickness occur in the eyes of patients with BRVO. One study used optical coherence tomography angiography (OCTA) to evaluate changes in peripapillary microvasculature parameters in the fellow eyes of the patients with unilateral RVO.[8] The study findings indicated that the mean RNFL and ganglion cell-inner plexiform layer thicknesses in the fellow eyes of RVO patients are significantly thinner, compared with normal control eyes. They also found that the peripapillary perfusion density is significantly lower compared with control group eyes. Taken together, these study results indicate that the changes in the peripapillary choroid and ONH are similar between patients with BRVO and patients with glaucoma.[3–8] Thus, our hypothesis was that BRVO and glaucoma may share a common pathophysiology that results in these similarities.

The LC is a fenestrated, three-dimensional mesh-like structure of connective tissue beams where the retinal ganglion cell (RGC) axons exit from the eye.[9] These load-bearing trabeculae, many of which contain capillaries, provide structural support and a nutrient supply to the RGC axons.[9] If laminar tissue is damaged or lost, the RGC axons may lose this structural and functional support. RGC axon compression, extension, shearing, or impaired nutrient delivery, or some combination of these changes, can affect the axons. The presence of LC defects is associated with normal-tension glaucoma (NTG) than open-angle glaucoma, more advanced glaucoma stage, and glaucomatous visual field progression.[10–14] Although the changes in peripapillary choroid and LC seem to be similar between patients with BRVO and patients with glaucoma, no study has compared the prevalence of focal LC defects between these two groups of patients.

In this study, we investigated the prevalence of focal LC defects in patients with unilateral BRVO. We also compared ocular characteristics between the eyes with and without focal LC

defects in these patients. In addition, we compared ocular characteristics among the eyes with focal LC defects and BRVO, those with focal LC defects and NTG, and the control group.

## Methods

### Enrollment of study population

This retrospective study was performed at the Catholic Kwandong University College of Medicine, International St. Mary's Hospital, Incheon, Republic of Korea. The study protocol was approved by the Institutional Review Board of International St. Mary's Hospital, Catholic Kwandong University and adhered to all tenets of the Declaration of Helsinki. The requirement for informed consent from each patient was waived due to the retrospective nature of the investigation. This waiver was approved by the Institutional Review Board of the International St. Mary's Hospital.

We performed a retrospective review of medical records from April 2019 to August 2019 and selected patients for inclusion in the study groups. Group 1 was defined as the patients with unilateral BRVO (BRVO group). Group 2 was defined as the patients with NTG (Glaucoma group). Group 3 was the patients who visited the ophthalmology clinic for vitreous floaters, but had no other concomitant ocular disease; these patients underwent a thorough ophthalmologic evaluation (Control group). Based on the results of our previous studies, [3,4,15] ONH evaluation is currently included in the routine ophthalmologic evaluation for patients with vitreoretinal disease. This inclusion of the ONH evaluation enabled us to perform this study.

The inclusion criteria were: 1) No concomitant ocular disease such as diabetic retinopathy, age-related macular degeneration, or ocular ischemic syndrome; 2) No significant media opacity such as vitreous opacity or advanced cataract that negatively affected the accuracy of optical coherence tomography (OCT) imaging of the retina, ONH, or choroidal structures; and 3) eyes with any sign of pathologic myopia such as fundus changes indicative of pathologic myopia, including lacquer cracks, atrophic patches, or chorioretinal atrophy. NTG was defined as evidence of: 1) a glaucomatous visual field defect, confirmed by two reliable visual field tests; 2) a glaucomatous ONH, indicated by a cup-disc ratio >7 and a cup-disc ratio asymmetry >0.2 with diffuse or focal neuroretinal rim thinning, disc hemorrhage, or vertical elongation of the optic cup, or some combination of these clinical signs; and intraocular pressure (IOP) less than 21 mmHg by both Goldmann contact tonometer and non-contact tonometer.

The primary outcome measure was the prevalence of focal LC defect in the patients with unilateral BRVO. In addition, we compared ocular characteristics, including mean PCT values, between the eyes with and without focal LC defect. We also compared the ocular characteristics of the eyes with focal LC defect among the patients in the BRVO, glaucoma, and control groups.

### Ophthalmologic examination

Detailed ophthalmologic examination was performed for each patient in the study population according to the routine retinochoroidal evaluation used in our clinic, as previously described. [3,4,15] The routine ophthalmologic evaluation included a slit lamp examination, IOP measurement using a non-contact tonometer, and a fundus examination after dilation. An autorefractor was used to measure the refractive error for each eye; the result was converted to spherical equivalents [diopters (D)]. The evaluation of the patients with unilateral BRVO included fluorescein angiography, fundus auto fluorescence, and spectral domain OCT (Spectralis; Heidelberg Engineering, Heidelberg, Germany) with an enhanced depth imaging modality. The fluorescein angiography was performed using the Heidelberg Retina

Angiograph system (HRA-2; Heidelberg Engineering) equipped with a confocal scanning laser ophthalmoscope.

## Optic nerve head and retinal evaluation using spectral domain optical coherence tomography

For the LC analysis, serial horizontal cross-sectional scans that were approximately 30 μm apart and that included the entire ONH were performed. The enhanced depth imaging OCT images of the ONH were reviewed by two specialized investigators (HMK and EWK) who were blinded to each patient's information. If the two reviewers disagreed about whether a focal LC defect was present, detailed discussions were used to make the final determination. Focal LC defect was defined as an anterior laminar surface irregularity violating the normal smooth curvilinear U- or W-shaped contour. To avoid a false positive result, each defect needed to be >100 μm in diameter, >30 μm in depth, and present in at least one additional raster scan (Fig 1A).[16]

The choroidal thickness (CT) was defined as the perpendicular distance from the outer border of the hyperreflective line corresponding to the retinal pigment epithelium to chorio-scleral interface.[3,4,15] For measurement of the PCT, a circular scan was centered on the

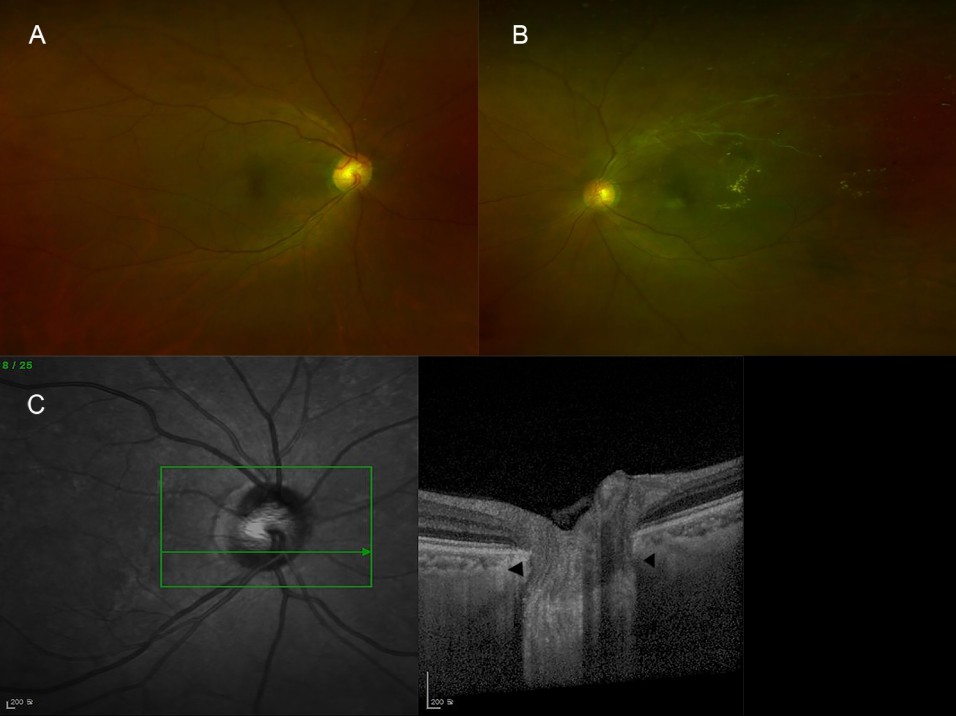

**Fig 1. A representative image showing optic nerve head (ONH), peripapillary choroidal thickness (PCT), and subfoveal choroidal thickness (SFCT) assessment using spectral domain optical coherence tomography equipped with an enhanced depth imaging modality.** (A) For the ONH assessment, serial horizontal cross-sectional scans that were approximately 30 μm apart and that included the entire ONH were performed. (B) For measurement of PCT, a circular scan was centered on the ONH (3.40-mm diameter, RNFL circle scan) in the peripapillary area. The PCT was measured at eight points [arrowheads: superior (S), superonasal (SN), nasal (N), inferonasal (IN), inferior (I), inferotemporal (IT), temporal (T), and superotemporal (ST)]. Average PCT values were calculated. (C) Choroidal thickness was defined as the perpendicular distance from the outer border of the hyperreflective line representing the retinal pigment epithelium (upper arrowhead) to the chorio-scleral interface (lower arrowhead). Using digital calipers, the SFCT was measured at the subfoveal region in both the horizontal and vertical images; the two measurements were averaged to obtain the mean SFCT value.

ONH [3.40-mm diameter, RNFL circle scan] in the peripapillary area.[3,4,15] Using the digital calipers provided by the Heidelberg Spectralis OCT software, the PCT was measured at eight points (superior, superonasal, nasal, inferonasal, inferior, inferotemporal, temporal, and super-otemporal; Fig 1B) and then the mean PCT value was calculated.

For the macular chorioretinal evaluation, serial cross-sectional horizontal scans approximately 121 μm apart in a 30˚x15˚ macular area were obtained.[3,4,15] Separately-obtained single horizontal and single vertical scans across the fovea were also performed. To measure the subfoveal choroidal thickness (SFCT), at least two good-quality horizontal and vertical scans across the fovea were performed for each eye. Using the digital calipers provided by the Heidelberg Spectralis OCT software, the CT was measured horizontally and vertically at the subfoveal region in each trans-sectional image, and the average value was calculated (Fig 1C). The PCT and SFCT were measured by two independent observers (HMK and JHC) who were blinded to the clinical data for each patient.

The central macular thickness (CMT) was defined as a mean retinal thickness in the central subfield with a diameter of 1.0 mm around the fovea. The inner and the outer rings had diameters of 3.0 mm and 6.0 mm, respectively, and the CMT was automatically calculated by the embedded program in the spectral domain OCT software.

RNFL thickness was automatically measured using the RNFL circle scan included with the Spectralis software, as described above.[3,4,15] RNFL thickness was measured in six sectors (temporal, superotemporal, superonasal, nasal, inferonasal, and inferotemporal), and the global RNFL thickness was automatically calculated by the software.

## Statistical analysis

The results were presented as mean ± standard deviation values, unless otherwise indicated. The IBM SPSS Statistics Version 22.0 software for Windows (IBM Corporation, Somers, NY, USA) was used for the statistical analyses. Mauchly's test of sphericity and Kolmogorov-Smirnov analyses were used to confirm statistical validity. Ocular characteristics including refractive errors, axial length, the mean SFCT, the mean PCT, the mean global RNFL thickness, and the mean CMT were compared among the study groups. The Mann-Whitney U test for continuous variables and the chi-square test for categorical valuables were used to compare two groups. The Kruskal-Wallis test was used to compare continuous variables among the three groups. The results from the right eyes were chosen for statistical analysis of the control group and bilateral LC defects patients. Results with a P<0.05 were considered statistically significant.

## Results

### Baseline characteristics of study population

For this study, the data from 39 patients with unilateral BRVO, 36 patients with NTG, and 46 patients in the control group were retrospectively analyzed. The mean age at the time of examination was 63.4±11.3 years in the patients with unilateral BRVO, 56.0±12.0 years in the patients with NTG, and 61.7±1.6 years in the control group (P = 0.018). Focal LC defects were detected in 20 eyes of 14 patients (38.9%) in the patients with unilateral BRVO, 24 eyes of 15 patients (41.7%) in the patients with NTG; no focal LC defects were detected in the eyes of the control group patients (P<0.001).

### Lamina cribrosa defects in the patients with normal-tension glaucoma

Among the patients in the Glaucoma group, 24 eyes of 15 patients (41.7%) showed LC defects. Nine patients had bilateral focal LC defects, and six patients had unilateral focal LC defects.

**Table 1. Comparison of ocular characteristics between eyes with focal lamina cribrosa (LC) defects and those without in patients with glaucoma.**

| | The eyes with LC defects (24 eyes, 15 patients) | The eyes without LC defects (48 eyes, 21 patients) | P value |
|---|---|---|---|
| Cup-disc ratio | 0.6±0.1 | 0.6±0.1 | 0.373 |
| Mean RE (diopters) | -1.3±2.2 | -0.9±2.0 | 0.541 |
| Mean AXL (mm) | 24.2±1.0 | 23.9±1.3 | 0.362 |
| Mean IOP (mmHg) | 13.2±3.0 | 12.9±3.0 | 0.653 |
| Mean CMT (μm) | 263.2±25.8 | 265.9±19.7 | 0.631 |
| Mean SFCT (μm) | 217.6±75.2 | 257.7±90.9 | 0.066 |
| Mean global RNFL (μm) thickness | 89.9±16.1 | 93.3±15.9 | 0.570 |
| Mean PCT (μm) | 133.1±48.9 | 170.8±81.9 | 0.042 |

Abbreviations: AXL, axial length; CMT, central macular thickness; IOP, intraocular pressure; LC, lamina cribrosa; PCT, peripapillary choroidal thickness; RE, refractive error; RNFL, retinal nerve fiber layer; SFCT, subfoveal choroidal thickness

Statistical analysis was performed by *Mann-Whitney U test for the continuous variables. Results with P<0.05 were considered statistically significant.

We compared the ocular characteristics between the eyes with and without focal LC defects, and the mean PCT was significantly thinner in the eyes with focal LC defect than in those without (133.1±48.9 μm versus 170.8±81.9 μm, respectively; P = 0.042). The other factors were not significantly different between the eyes with and without focal LC defects (Table 1).

## Lamina cribrosa defects in the patients with unilateral branch retinal vein occlusion

Twenty eyes of 14 patients (38.9%) in the BRVO group had LC defects. Among the 20 eyes with LC defects, 10 eyes were affected by BRVO. The other 10 eyes were unaffected fellow eyes. In the BRVO-affected eyes, the mean PCT was 102.7±31.1 μm in the eyes with focal LC defects and 163.1±70.1 μm in the eyes without LC defects (P = 0.009). The other ocular characteristics were not significantly different between the eyes with and without focal LC defects (Table 2).

In the unaffected fellow eyes, the mean PCT was 114.0±56.2 μm in the eyes with focal LC defect, and 170.6±60.9 μm in the defect-free eyes (P = 0.014). The other ocular characteristics were not significantly different between the eyes with and without focal LC defects (Table 3).

## Comparison among the patients with unilateral branch retinal vein occlusion and lamina cribrosa defects, glaucoma and lamina cribrosa defects, and the normal control group

We compared ocular characteristics among the eyes with focal LC defect in the BRVO group, the glaucoma group, and the control groups. Among the various ocular characteristics, the mean cup-disc ratio was also significantly different among the three groups: 0.5±0.1 in the BRVO group, 0.6±0.1 in the glaucoma group, and 0.4±0.1 in the control group (P<0.001). The values for the mean PCT, the mean SFCT, and the mean global RNFL thickness were also significantly different among the groups (P = 0.01, 0.046, and P = 0.039; respectively). There were no statistically significant differences in the other ocular characteristics among three groups (Table 4).

Representative figures are shown as Figs 2 and 3.

**Table 2. Comparison of ocular characteristics between eyes with focal lamina cribrosa (LC) defects and those without in eyes affected by branch retinal vein occlusion.**

|  | The eyes with LC defects (10 eyes) | The eyes without LC defects (29 eyes) | P value |
|---|---|---|---|
| Cup-disc ratio | 0.4±0.1 | 0.4±0.1 | 0.775 |
| Mean RE (Diopters) | 2.1±1.1 | -0.2±0.8 | 0.134 |
| Mean AXL (mm) | 23.4±0.4 | 23.8±1.3 | 0.507 |
| Mean IOP (mmHg) | 11.9±2.0 | 12.8±2.8 | 0.368 |
| Mean CMT (μm) | 277.1±57.3 | 300.6±89.5 | 0.421 |
| Mean SFCT (μm) | 214.0±84.3 | 242.1±70.0 | 0.281 |
| Mean global RNFL thickness (μm) | 98.5±18.5 | 106.3±20.0 | 0.265 |
| Mean PCT (μm) | 102.7±31.1 | 163.1±70.1 | 0.009 |

Abbreviations: AXL, axial length; CMT, central macular thickness; IOP, intraocular pressure; LC, lamina cribrosa; PCT, peripapillary choroidal thickness; RE, refractive error; RNFL, retinal nerve fiber layer; SFCT, subfoveal choroidal thickness

Statistical analysis was performed by Mann-Whitney U test for the continuous variables. Results with P<0.05 were considered statistically significant.

## Discussion

In this study, we investigated the prevalence of foal LC defects in patients with unilateral BRVO. We also compared ocular characteristics between eyes with and without focal LC defects. Approximately 39% of the patients with unilateral BRVO had focal LC defects. LC defects were present in both BRVO-affected eyes and the unaffected fellow eyes. The mean PCT was significantly thinner in the eyes with focal LC defects than those without. The mean SFCT was also thinner in the eyes with focal LC defect than in those without, but the difference was not statistically significant. The mean PCT was significantly thinner in the eyes with focal LC defects than in the eyes without in both the BRVO group and the glaucoma group. We

**Table 3. Comparison of ocular characteristics between eyes with lamina cribrosa (LC) defects and those without in the unaffected fellow eyes in patients with branch retinal vein occlusion.**

|  | The eyes with LC defects (10 eyes) | The eyes without LC defects (29 eyes) | P value |
|---|---|---|---|
| Cup-disc ratio | 0.5±0.1 | 0.5±0.1 | 0.588 |
| Mean RE (Diopters) | -0.9±1.3 | 1.0±2.1 | 0.295 |
| Mean AXL (mm) | 23.6±0.3 | 23.7±1.2 | 0.906 |
| Mean IOP (mmHg) | 13.7±2.5 | 12.9±3.0 | 0.473 |
| Mean CMT (μm) | 279.2±80.6 | 275.8±35.8 | 0.855 |
| Mean SFCT (μm) | 215.1±72.2 | 264.0±87.0 | 0.167 |
| Mean global RNFL thickness (μm) | 88.5±15.8 | 97.1±14.7 | 0.125 |
| Mean PCT (μm) | 114.0±56.2 | 170.6±60.9 | 0.014 |

Abbreviations: AXL, axial length; CMT, central macular thickness; IOP, intraocular pressure; LC, lamina cribrosa; PCT, peripapillary choroidal thickness; RE, refractive error; RNFL, retinal nerve fiber layer; SFCT, subfoveal choroidal thickness

Statistical analysis was performed by Mann-Whitney U test for the continuous variables. Results with P<0.05 were considered statistically significant.

**Table 4. Comparison of baseline characteristics among eyes with lamina cribrosa (LC) disinsertion in patients with branch retinal vein occlusion (BRVO), eyes with LC disinsertion in patients with glaucoma, and the control group.**

|  | LC disinsertion in BRVO (20 eyes, 14 patients) | LC disinsertion in glaucoma (24 eyes, 15 patients) | Control group (46 eyes, 46 patients) | P value |
|---|---|---|---|---|
| Mean age | 67.8±8.6 | 57.3±7.6 | 61.7±1.6 | 0.018 |
| Cup-disc ratio | 0.5±0.1 | 0.6±0.1 | 0.3±0.1 | <0.001 |
| Mean RE (Diopters) | 0.8±1.8 | -1.3±1.2 | -0.2±1.0 | 0.181 |
| Mean AXL (mm) | 23.5±0.4 | 24.2±1.0 | 23.8±0.8 | 0.139 |
| Mean IOP (mmHg) | 12.5±2.0 | 13.2±3.0 | 12.8±1.8 | 0.710 |
| Mean CMT (μm) | 285.5±66.7 | 263.2±25.8 | 260.7±21.7 | 0.202 |
| Mean SFCT (μm) | 202.6±79.6 | 217.6±75.2 | 251.2±85.2 | 0.046 |
| Mean global RNFL thickness (μm) | 95.3±17.7 | 89.9±16.1 | 102.2±9.9 | 0.039 |
| Mean PCT (μm) | 111.4±45.1 | 133.1±48.9 | 158.2±50.1 | 0.006 |

Abbreviations: AXL, axial length; CMT, central macular thickness; IOP, intraocular pressure; LC, lamina cribrosa; PCT, peripapillary choroidal thickness; RE, refractive error; RNFL, retinal nerve fiber layer; SFCT, subfoveal choroidal thickness

Statistical analysis was performed using the Kruskal-Wallis test for the continuous variables. Results with P<0.05 were considered statistically significant.

found that there were no obvious focal LC defects in the control group eyes, and the mean PCT was significantly thicker than in the eyes with focal LC defects in the BRVO group and the glaucoma group. Our findings were consistent with previous findings, which indicated that there are similarities in the changes in the parapapillary area and ONH between patients with BRVO and glaucoma.[3–8] The finding of significant thinning of the mean PCT in the eyes with focal LC defects in both the BRVO and glaucoma groups supports the hypothesis that vascular insufficiency is, to some extent, involved in the pathophysiology of these two diseases.

The short posterior ciliary artery (SPCA) supplies blood to the posterior choriocapillaries, from the macular to the equator and the peripapillary areas.[17] Much of the perfusion of the tissues associated with the LC, including the RGC axons, is believed to be through microvessels and capillaries from the SPCA, which enters the sclera canal from the periphery at the LC level of insertion.[18–21] Because laminal capillaries have no collateral blood supply,[22,23] any biomechanical disturbance of blood supply due to laminar tissue damage or loss leads to a loss of structural and functional support to the RGC axons.

Study findings based on OCT angiography suggest that microvasculature dropout (MvD) is associated with focal LC defects in patients with glaucoma; these findings support the hypothesis that vascular insufficiency is present in these eyes.[24–29] MvD is a circumferential area of capillary dropout; its border adjoins the optic disc margin.[24–29] These studies found that the location of MvD is topographically associated with focal LC defect locations and that both focal LC defects and MvD are associated with peripapillary atrophy in glaucomatous eyes. [24–29] The findings of one study also suggested that compared with focal LC defects, the presence of MvD is more strongly associated with glaucoma severity.[26] This study also found that MvD was not, but focal LC defects were, detected the normal control group.[26] Another study investigated the topographic correlation between juxtapapillary PCT and MvD.[30] Localized thinning of the mean juxtapapillary PCT value was observed at the location of MvD in the primary open-angle glaucoma eyes with a parapapillary γ-zone.[30] Taken together, these findings suggest that there is a correlation between MvD and local LC defects in patients with glaucoma, but the details of the associated pathophysiologic process remains poorly understood.

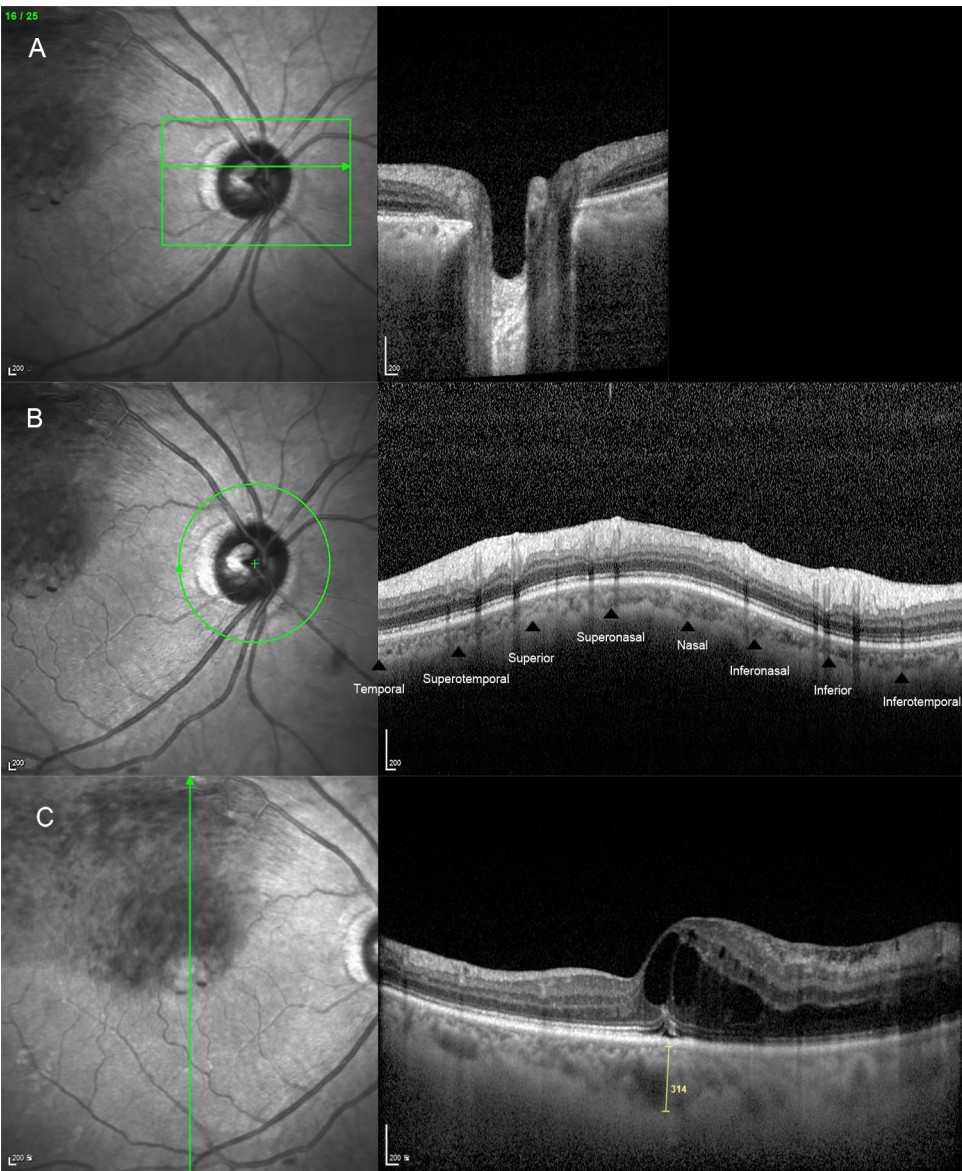

**Fig 2. A representative figure of a 59-year-old female patient who was diagnosed as branch retinal vein occlusion in the left eye.** (A) No specific sign in the right eye. (B) Ghost vessel at superotemporal vascular arcade with hard exudates and some retinal hemorrhages in the left eye. (C) Lamina cribrosa defect in the right eye (arrow head).

We lacked OCT angiography data in this study. However, significant thinning of the mean PCT in the eyes with focal LC defect, compared with those without, may indirectly reflect more vascular insufficiency in the eyes with focal LC defect. The results for the patients with BRVO and the patients with glaucoma showed a similar tendency. This finding suggested that in the BRVO group eyes, the parapapillary area and the ONH underwent pathophysiologic changes similar to those of the glaucoma patients. Systemic vascular diseases associated with thickening of the retinal artery, such as hypertension and arteriosclerosis, are risk factors for BRVO[31–34] and for glaucoma.[35–38] These systemic conditions can lead to increased resistance to flow, reduced perfusion pressure, increased blood viscosity, or some combination of these changes. The resulting vascular insufficiency of the ONH results in various changes

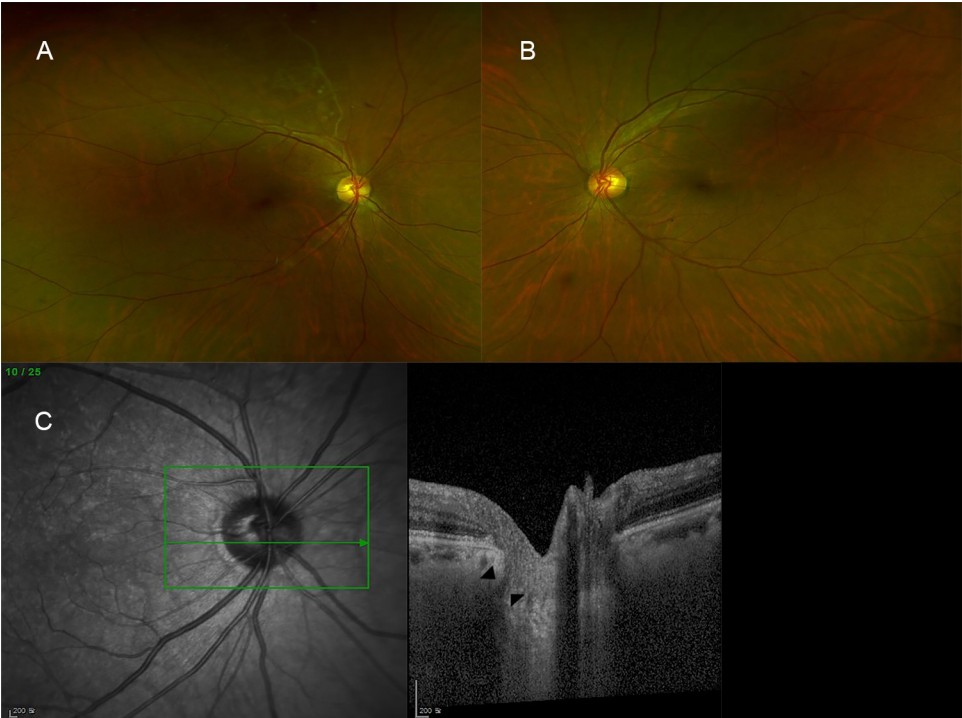

**Fig 3. A representative figure of a 64-year-old female who was diagnosed as branch retinal vein occlusion in the right eye.** (A) Ghost vessels at superior periphery in the right eye. Vitreous hemorrhage developed in the right eye after 2 months. (B) No specific sign in the left eye. (C) Lamina cribrosa defect in the right eye (arrow head).

such as peripapillary choroidal thinning and the development of MvD. Thus, vascular insufficiency in the parapillary area and at the periphery of LC insertion may result in a reduced blood flow in the laminar beams and negatively affect the nutrient supply to the RGC axons. If both BRVO and glaucoma share a common pathophysiologic change that affects the parapillary area and the ONH, the findings of previous studies can also be explained more easily. [3–8] However, further prospective studies are needed to determine the specific pathophysiologic mechanism associated with peripapillary choroidal change and focal LC defects in patients with BRVO.

In addition to significant thinning of the mean PCT, we found that the mean SFCT tended to be thinner in the eyes with focal LC defect than in those without in both the BRVO and glaucoma patients. Although the difference was not statistically significant, the thinning of the SFCT may be correlated with thinning of the PCT in these eyes. Because the macular choroid and peripapillary choroid receive blood flow from the SPCA,[16] chronic vascular insufficiency or disturbance of blood flow may concomitantly result in both macular and peripapillary choroidal thinning. Because the SPCA enters the scleral canal from the periphery at the level of LC insertion,[17–21] any biomechanical change at the LC insertion area may affect blood flow to branches of the SPCA. Or, vascular insufficiency to the SPCA itself may lead to these results. Vascular insufficiency to the SPCA is associated with systemic risk factors such as hypertension or atherosclerosis in both BRVO and glaucoma patients.[31–38]

Although we found a relatively higher prevalence of focal LC defect and associated PCT thinning in the patients with BRVO, the clinical effects of focal LC defect in these patients should be further investigated because visual field test results were not available for this study population. One study investigated glaucoma progression in eyes with glaucoma that

developed unilateral BRVO in the contralateral eyes.[39] The study results indicated that BRVO occurred in the eyes with more advanced-stage glaucoma, and the patients who developed BRVO had more rapid glaucoma progression in the contralateral eyes without BRVO, compared with the patients with glaucoma who did not develop BRVO.[39] Development of BRVO can add more vascular insults to the eyes of glaucoma patients and can result in further deterioration of visual function. Future studies should investigate the clinical effects of parapapillary choroidal changes and of ONH changes on glaucomatous visual field progression in patients with BRVO. If there is a more 'vulnerable ONH' appearance among patients with BRVO that is indicated by focal LC defects or other glaucomatous changes, more clinical attention should be given to the patients at risk to prevent further visual loss due to the effects of the BRVO and the associated glaucoma. It is also another possibility that the presence of focal LC defect is simply anatomical variation in patients with BRVO, because focal LC defects can be found in normal control group eyes.[26] If there are no significant effects of focal LC defects in patients with BRVO, physicians could reduce the intensity of monitoring for glaucoma in patients with BRVO. Thus, prospective, longitudinal studies should be performed to investigate the clinical effects of focal LC defects in patients with BRVO.

This study had several limitations, including use of a relatively small study population and a retrospective design. We also lacked data for various factors that are commonly considered in glaucoma studies, such as central corneal thickness and intraocular perfusion pressure, because these parameters are not routinely performed for patients with retinal disease. The use of 48 radial line B-scans centered on the ONH seems to be more applicable when investigating focal LC defects because the results provide clearer location information and spatial correlation with other parameters. Thus, further prospective studies that account for these limitations are warranted to validate our findings. Additional future studies on MvD in patients with BRVO will be also needed to investigate whether there is an association between MvD and PCT changes and the effects of MvD on glaucoma progression in these patients.

In conclusion, about 40% of the patients with unilateral BRVO had focal LC defects in BRVO-affected eyes and unaffected fellow eyes. The eyes with focal LC defects had significantly thinner mean PCT values than those without in the patients with BRVO and the patients with glaucoma. The eyes with focal LC defect in the patients with BRVO and the patients with glaucoma also had significantly thinner mean PCT values than those in the control group, suggesting possible pathyphysiologic correlation between BRVO and glaucoma.

## Author Contributions

**Conceptualization:** Hae Min Kang, Jeong Hoon Choi, Hyoung Jun Koh, Sung Chul Lee.

**Data curation:** Hae Min Kang, Eun Woo Kim, Jeong Hoon Choi.

**Formal analysis:** Hae Min Kang, Eun Woo Kim, Jeong Hoon Choi, Sung Chul Lee.

**Funding acquisition:** Hae Min Kang.

**Investigation:** Hae Min Kang, Eun Woo Kim, Jeong Hoon Choi, Hyoung Jun Koh, Sung Chul Lee.

**Methodology:** Hae Min Kang, Eun Woo Kim, Jeong Hoon Choi.

**Project administration:** Hae Min Kang, Hyoung Jun Koh, Sung Chul Lee.

**Resources:** Hae Min Kang, Hyoung Jun Koh.

**Software:** Hae Min Kang.

**Supervision:** Hae Min Kang, Jeong Hoon Choi, Hyoung Jun Koh, Sung Chul Lee.

**Validation:** Hae Min Kang, Eun Woo Kim, Jeong Hoon Choi, Sung Chul Lee.

**Visualization:** Hae Min Kang, Hyoung Jun Koh.

**Writing – original draft:** Hae Min Kang.

**Writing – review & editing:** Hae Min Kang, Eun Woo Kim, Jeong Hoon Choi, Hyoung Jun Koh, Sung Chul Lee.

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
