## [Decision Letter · Decision Letter 0]

20 Jan 2020

PONE-D-19-34472

Focal lamina cribrosa defect and significant peripapillary choroidal thinning in patients with unilateral branch retinal vein occlusion

PLOS ONE

Dear Prof Kang,

Thank you for submitting your manuscript to PLOS ONE. After careful consideration, we feel that it has merit but does not fully meet PLOS ONE’s publication criteria as it currently stands. Therefore, we invite you to submit a revised version of the manuscript that addresses the points raised during the review process.

The MS was examined by the reviewer and was found to have areas in need of improvement. Please revise accordingly and resubmit. Please understand that the revised MS may be seen by other reviewers as well. 

We would appreciate receiving your revised manuscript by Mar 05 2020 11:59PM. To enhance the reproducibility of your results, we recommend that if applicable you deposit your laboratory protocols in protocols.io, where a protocol can be assigned its own identifier (DOI) such that it can be cited independently in the future. For instructions see: http://journals.plos.org/plosone/s/submission-guidelines#loc-laboratory-protocols

We look forward to receiving your revised manuscript.

Kind regards,

Demetrios G. Vavvas

Academic Editor

PLOS ONE

Journal Requirements:

This work was supported by a National Research Foundation of Korea (NRF) grant funded by the Korean government (MSIT) (No.2018R1C1B5085620).

by the Korea government (MSIT) (No. 2018R1C1B5085620). 

None

We note that one or more of the authors are employed by a commercial company: Choikang Seoule Eye Clinic

Reviewers' comments:

Reviewer's Responses to Questions

**Comments to the Author**

1. Is the manuscript technically sound, and do the data support the conclusions?

Reviewer #1: Yes

2. Has the statistical analysis been performed appropriately and rigorously? 

Reviewer #1: Yes

3. Have the authors made all data underlying the findings in their manuscript fully available?

Reviewer #1: No

4. Is the manuscript presented in an intelligible fashion and written in standard English?

Reviewer #1: No

5. Review Comments to the Author

Reviewer #1: 1. The manuscript technically sound, the data do support the conclusions.

2. Statistical analysis is adequately described and scientifically sound

3. Authors state that some restrictions apply on the data presented in the manuscript

4. Multiple errors in English language were found including:

Title: should be corrected to read 'defects' instead of 'defect'

Abstract's purpose: 'defects' instead of 'defect'

Abstract conclusion:the phrase 'regardless of BRVO affection' is incorrect.

Introduction: 'Whether' instead of 'where', 'similar to that described in central retinal vein occlusion and hemi-central retinal vein occlusion.5' instead of 'like central retinal vein occlusion and hemi-central retinal vein occlusion.5'

Discussion: 'It is also another possibility that the presence' instead of 'It is also another possible that the presence'

Conclusions: the phrase 'regardless of BRVO affection' is incorrect.

5. Other comments:

a.The study included 3 study groups: BRVO, NTG and control group. The NTG group is selectively not mentioned in sections like the title and the abstract's purpose & conclusion.

b.Abstract methods could be more specific interns of parameters studied.

c.The main hypothesis described at the end of the introduction is interesting, yet it is not mentioned anywhere in the abstract nor in the manuscripts conclusion. 'our hypothesis was that BRVO and glaucoma may share a common pathophysiology that results in these similarities.'

d.Only one of the Representative figures 2,3,4 is needed.

e.Funder's role in the study is not specified. If the funder played any role in study design, data collection and analysis, decision to publish, or preparation of the manuscript that should be stated; otherwise please include this sentence at the end of your statement: 'The funders had no role in study design, data collection and analysis, decision to publish, or preparation of the manuscript.'

6. PLOS authors have the option to publish the peer review history of their article (what does this mean?). If published, this will include your full peer review and any attached files.

Reviewer #1: No

---

## [Author Response · Author response to Decision Letter 0]

31 Jan 2020

On behalf of all authors of the manuscript, I wish to thank you and all the reviewers for their valuable comments and constructive input into our manuscript PONE-D-19-34472 ‘Focal lamina cribrosa defect and significant peripapillary choroidal thinning in patients with unilateral branch retinal vein occlusion’. We have tried our best to address all critiques and made all suggested additions and changes. The followings are the answers to the reviewers’ comments.

Reviewer comments

Reviewer #1: 

The manuscript technically sound, the data do support the conclusions.

Statistical analysis is adequately described and scientifically sound

Authors state that some restrictions apply on the data presented in the manuscript

Multiple errors in English language were found including:

Title: should be corrected to read 'defects' instead of 'defect'

Abstract's purpose: 'defects' instead of 'defect'

Abstract conclusion:the phrase 'regardless of BRVO affection' is incorrect.

Introduction: 'Whether' instead of 'where', 'similar to that described in central retinal vein occlusion and hemi-central retinal vein occlusion.5' instead of 'like central retinal vein occlusion and hemi-central retinal vein occlusion.5'

Discussion: 'It is also another possibility that the presence' instead of 'It is also another possible that the presence'

Conclusions: the phrase 'regardless of BRVO affection' is incorrect.

Answer: Thank you for the comments, and we authors apologize for the linguistic errors. We reviewed the manuscript, and corrected multiple linguistic errors as commented. 

5. Other comments:

a.The study included 3 study groups: BRVO, NTG and control group. The NTG group is selectively not mentioned in sections like the title and the abstract's purpose & conclusion.

Answer: Thank you for the comment. The NTG group was commented in the abstract and conclusion, however, we also agree that the reviewers and potential readers may have similar opinions. Because our primary endpoint is focused on the prevalence of LC defect and associated ocular characteristics in the patients with BRVO, these are mainly described in the limited spaces of ABSTRACT section or CONCLUSION section. The NTG group and the normal control group were included for comparison with BRVO group, and we authors think that we fully described in the RESULT section and DISCUSSION section. 

b. Abstract methods could be more specific interns of parameters studied.

Answer: Thank you for the comment, and we added more details in the ABSTRACT section. 

c.The main hypothesis described at the end of the introduction is interesting, yet it is not mentioned anywhere in the abstract nor in the manuscripts conclusion. 'our hypothesis was that BRVO and glaucoma may share a common pathophysiology that results in these similarities.'

Answer: Thank you for the comment, and we authors agree with the comment. We authors added this in the ABSTRACT section and CONCLUSION section. 

d.Only one of the Representative figures 2,3,4 is needed.

Answer: Thank you for the comment, and we authors discussed this. We authors intended to present various cases, however, we also agreed that these cases may be too much for the potential readers. However, if it is possible, we authors would like to present two figures, presenting both focal LC defects in BRVO-affected eye and unaffected fellow eye. If it is not acceptable due to various reasons such as limitation of space, we further remove one figure. We appreciate again for this comment. 

e.Funder's role in the study is not specified. If the funder played any role in study design, data collection and analysis, decision to publish, or preparation of the manuscript that should be stated; otherwise please include this sentence at the end of your statement: 'The funders had no role in study design, data collection and analysis, decision to publish, or preparation of the manuscript.'

Answer: Thank you for the comment. Because the Funder of this study, the National Rearch Foundation of Korea had no role in the study design, data collection and analysis, decision to publish, or preparation of the manuscript, we added this in the manuscript. As recommended by the Journal, we added this to the Funding Statement section. 

We believe that the revised manuscript would be much better than the previous one. The changed parts and sentences are marked on the 'answers' in each comment. We hope this revised manuscript would meet the requirements of the Retina for publication.

---

## [Decision Letter · Decision Letter 1]

26 Feb 2020

Focal lamina cribrosa defects and significant peripapillary choroidal thinning in patients with unilateral branch retinal vein occlusion

PONE-D-19-34472R1

Dear Dr. Kang,

We are pleased to inform you that your manuscript has been judged scientifically suitable for publication and will be formally accepted for publication once it complies with all outstanding technical requirements.

With kind regards,

Demetrios G. Vavvas

Academic Editor

PLOS ONE

Additional Editor Comments (optional):

Reviewers' comments:

Reviewer's Responses to Questions

**Comments to the Author**

1. If the authors have adequately addressed your comments raised in a previous round of review and you feel that this manuscript is now acceptable for publication, you may indicate that here to bypass the “Comments to the Author” section, enter your conflict of interest statement in the “Confidential to Editor” section, and submit your "Accept" recommendation.

Reviewer #1: All comments have been addressed

2. Is the manuscript technically sound, and do the data support the conclusions?

Reviewer #1: Yes

3. Has the statistical analysis been performed appropriately and rigorously? 

Reviewer #1: Yes

4. Have the authors made all data underlying the findings in their manuscript fully available?

Reviewer #1: No

5. Is the manuscript presented in an intelligible fashion and written in standard English?

Reviewer #1: Yes

6. Review Comments to the Author

Reviewer #1: (No Response)

7. PLOS authors have the option to publish the peer review history of their article (what does this mean?). If published, this will include your full peer review and any attached files.

Reviewer #1: No

---

## [Editor Report · Acceptance letter]

2 Mar 2020

PONE-D-19-34472R1 

Focal lamina cribrosa defects and significant peripapillary choroidal thinning in patients with unilateral branch retinal vein occlusion 

Dear Dr. Kang:

I am pleased to inform you that your manuscript has been deemed suitable for publication in PLOS ONE. Congratulations! Your manuscript is now with our production department. 

With kind regards,

on behalf of

Dr. Demetrios G. Vavvas 

Academic Editor

PLOS ONE